# A dePEGylated Lipopeptide-Based Pan-Coronavirus Fusion Inhibitor Exhibits Potent and Broad-Spectrum Anti-HIV-1 Activity without Eliciting Anti-PEG Antibodies

**DOI:** 10.3390/ijms24119779

**Published:** 2023-06-05

**Authors:** Ling Xu, Chao Wang, Wei Xu, Lixiao Xing, Jie Zhou, Jing Pu, Mingming Fu, Lu Lu, Shibo Jiang, Qian Wang

**Affiliations:** 1Key Laboratory of Medical Molecular Virology (MOE/NHC/CAMS), Shanghai Institute of Infectious Disease and Biosecurity, School of Basic Medical Sciences, Fudan University, Shanghai 200032, China; 21111010081@m.fudan.edu.cn (L.X.); xuwei11@fudan.edu.cn (W.X.); 20111010065@fudan.edu.cn (L.X.); 19211010046@fudan.edu.cn (J.Z.); 17111010015@fudan.edu.cn (J.P.); 17301010019@fudan.edu.cn (M.F.); lul@fudan.edu.cn (L.L.); 2State Key Laboratory of Toxicology and Medical Countermeasures, Beijing Institute of Pharmacology and Toxicology, Beijing 100850, China; chaow301@sina.com

**Keywords:** HIV-1, coronavirus, six-helix bundle, broad-spectrum fusion inhibitor

## Abstract

We previously identified a lipopeptide, EK1C4, by linking cholesterol to EK1, a pan-CoV fusion inhibitory peptide via a polyethylene glycol (PEG) linker, which showed potent pan-CoV fusion inhibitory activity. However, PEG can elicit antibodies to PEG in vivo, which will attenuate its antiviral activity. Therefore, we designed and synthesized a dePEGylated lipopeptide, EKL1C, by replacing the PEG linker in EK1C4 with a short peptide. Similar to EK1C4, EKL1C displayed potent inhibitory activity against severe acute respiratory syndrome coronavirus 2 (SARS-CoV-2) and other coronaviruses. In this study, we found that EKL1C also exhibited broad-spectrum fusion inhibitory activity against human immunodeficiency virus type 1 (HIV-1) infection by interacting with the N-terminal heptad repeat 1 (HR1) of viral gp41 to block six-helix bundle (6-HB) formation. These results suggest that HR1 is a common target for the development of broad-spectrum viral fusion inhibitors and EKL1C has potential clinical application as a candidate therapeutic or preventive agent against infection by coronavirus, HIV-1, and possibly other class I enveloped viruses.

## 1. Introduction

Class I enveloped viruses, such as human immunodeficiency virus type 1 (HIV-1), severe acute respiratory syndrome coronavirus 2 (SARS-CoV-2), and Influenza A virus (IAV), pose a huge threat to global public health security, economic development, and social stability. HIV-1 is still one of the most serious public health threats in the 21st century [1]. The coronavirus disease 2019 (COVID-19) pandemic caused bySARS-CoV-2 has now caused approximately 863 million infections and 6.8 million deaths [2] (https://www.worldometers.info/coronavirus/, accessed on 1 March 2022). Other highly pathogenic coronaviruses, such as severe acute respiratory syndrome coronavirus (SARS-CoV), middle east respiratory syndrome coronavirus (MERS-CoV), and emerging novel coronaviruses in the future, have equally serious impacts on human health. These statistics call for the urgent development of broad-spectrum antiviral drugs to combat current and future emerging and re-emerging viruses.

Human coronavirus (HCoV) and HIV-1 are both class I enveloped viruses with similar membrane fusion mechanisms [3]. Specifically, during membrane fusion, the N-terminal heptad repeat 1 (HR1) of fusion glycoproteins (gp41 of HIV-1 and S2 subunit of coronavirus), together with the antiparallel C-terminal heptad repeat 2 (HR2), can form a highly stable six-helix bundle (6-HB), bringing the viral and cell membranes together for fusion [4,5,6,7,8]. Since both HIV-1 and HCoV envelope proteins form 6-HB, it is reasonable to suggest that HR1 in the HCoV S2 protein and HIV-1 gp41 can serve as common targets for the design of broad-spectrum drugs against HcoVs and HIV-1.

Our previous study has demonstrated that a lipopeptide-based pan-CoV fusion inhibitor, EK1C4, can form 6-HB with HR1 peptides of HCoV and HIV-1, thus exhibiting potent inhibitory activity against infection by HCoV and HIV-1 [9]. To synthesize EK1C4, four polyethylene glycol (PEG) linkers were used to link EK1, a pan-CoV fusion inhibitory peptide, and cholesterol, and such linkage was found to improve the peptide’s inhibitory activity. However, PEG has both immunogenic and antigenic properties and can induce antibodies to PEG in vivo. This means that the anti-PEG antibodies in organisms may lead to rapid clearance of the PEGylated drugs and neutralization of the inhibitors’ biological activity, thereby limiting their therapeutic effectiveness [10,11]. Therefore, in a previous study, we designed a dePEGylated pan-CoV fusion inhibitor, EKL1C (Figure 1), by replacing the PEG linker in EK1C4 with a short peptide, while still retaining robust broad-spectrum anti-coronavirus inhibitory activity, making it a potential candidate for future drug development [12]. In this study, we detected the inhibitory activity of EKL1C against HIV-1 infection, including HIV-1 pseudoviruses, HIV-1 laboratory-adapted strains, HIV-1 clinical isolates, and enfuvirtide (also known as T20)-resistant HIV-1 strains. Our study shows that like the PEGylated lipopeptide EK1C4, the dePEGylated lipopeptide EKL1C could also effectively inhibit infection by divergent HIV-1 strains. However, EKL1C is expected to not induce anti-PEG antibodies. Therefore, it is safer and more stable than EK1C4, thus having better potential to be further developed as a broad-spectrum fusion-inhibitor-based therapeutic or prophylatic for the treatment and prevention of infection by HCoVs, HIV-1, and possibly other class I enveloped viruses.

## 2. Results

### 2.1. The dePEGylated Lipopeptide-Based Pan-CoV Fusion Inhibitor EKL1C Exhibited Potent Inhibitory Activity against a Broad Spectrum of Pseudotyped HIV-1 Strains

Our previous study has shown that the dePEGylated lipopeptide-based pan-CoV fusion inhibitor EKL1C is highly effective against infection from divergent HCoVs with IC_50_ (half maximal inhibitory concentration) ranging from 26 to 148 nM [12]. In this study, we detected the potential inhibitory activity of EKL1C, using the HIV fusion-inhibitor-based anti-HIV drug T20 as a control, against infection from HIV-1 pseudotyped with a panel of HIV-1 envelopes (Envs) of different subtypes (A, B, C, D, G, A/E, A/G, B/C) with different coreceptor usages (X4, R5) and tiers (1A, 1B, 2, 3). As shown in Table 1, EKL1C also had strong inhibitory activity against different subtypes of HIV-1 pseudoviruses, with IC_50_ ranging from 5.1 to 186.5 nM, similar to that of T20 with IC_50_ ranging from 6.5 to 394.2 nM. These results suggest that in addition to its potent anti-HCoV activity, EKL1C also exhibits broad-spectrum inhibitory activity against infection from divergent pseudotyped HIV-1, such as the anti-HIV drugT20, which is derived from the HIV-1 gp41 HR2 region.

### 2.2. EKL1C Exhibited Potency against a Broad Spectrum of HIV-1 Laboratory-Adapted Strains

Next, we assessed the inhibitory activity of EKL1, EKL0C, and EKL1C against infection from HIV-1 laboratory-adapted strains, HIV-1 IIIB (X4) and HIV-1 Bal (R5), using T20 peptide, cholesterol, and Br-cholesterol as controls. We found that EKL0C and EKL1C could effectively inhibit infection from HIV-1 laboratory-adapted strains while EKL1, cholesterol, and Br-cholesterol had no inhibitory activity against HIV-1 laboratory-adapted strains at concentration as high as 5000 nM. The IC_50_ of EKL0C, EKL1C, and T20 against HIV-1 IIIB infection was 246.6, 73.7, and 47.3 nM, respectively, while that of EKL0C, EKL1C, and T20 against Bal infection was 77.7, 39.7, and 11.7 nM, respectively (Figure 2A,B), confirming that EKL1C exhibits significant inhibitory activity against infection from HIV-1 laboratory-adapted strains. EKL1 derived from the S protein HR2 domain without cholesterol conjugation exhibited no inhibitory activity against HIV-1 laboratory-adapted strains. However, both dePEGylated lipopeptides EKL0C and EKL1C, which are also derived from the S protein HR2 domain but conjugated with cholesterol, exhibited potent but variable inhibitory activity against infection from HIV-1 laboratory-adapted strains, and the inhibitory activity of EKL1C against HIV-1 laboratory-adapted strains was generally higher than that of EKL0C, which is consistent with the inhibitory activity against HCoV infection [12].

### 2.3. EKL1C Exhibited Potency against a Broad Spectrum of HIV-1 Primary Isolates

We further determined the inhibitory activity of EKL1C against a panel of HIV-1 clinical isolates with different subtypes (A, B, D, O) and different tropisms (X4, R5). As expected, EKL1C also showed strong antiviral activity against HIV-1 clinical isolates with IC_50_ values of EKL1C in the range of 70.1 to 192.9 nM and those of T20 ranging from 26.2 to 52.6 nM (Table 2). Particularly, both EKL1C and T20 were more effective against J32228M4 than other isolates, possibly because its spike protein S2 domain contains a sequence that is preferably bound by EKL1C and T20. Although EKLIC had a lower efficacy than T20 against clinical isolates in this in vitro experiment, it is still highly effective against infection from HIV-1 primary isolates, indicating that EKL1C has a broad-spectrum inhibitory activity against divergent HIV-1 strains. Hopefully, this phenomenon can be verified in an in vivo study by using an animal model in the future.

### 2.4. EKL1C Exhibited Potency against a Broad Spectrum of T20-Resistant HIV-1 Strains

Subsequently, we tested the inhibitory activity of EKL1C against infection from T20-resistant HIV-1 strains. As shown in Table 3, EKL1C had potent inhibitory activity against infection from T20-resistant strains, with IC_50_ ranging from 9.4 to 97.9 nM, whereas T20 could not inhibit infection from the five T20-resistant strains at a concentration as high as 1000 nM.

### 2.5. EKL1C Inhibited HIV-1 Infection at the Early Stage of Viral Replication

Here, we examined how EKL1C exerts its anti-HIV-1 effects. To accomplish this, we used a cell washout to determine whether EKL1C acts on host cells or not. As shown in Figure 3A, EKL1C completely inhibited HIV-1 IIIB infection when EKL1C-pretreated MT-2 cells were not washed. However, its inhibitory activity was significantly decreased after washing cells, but before adding the virus. These results suggest that the inhibitory activity of EKL1C is because of its interaction with the cell surface proteins, such as cell receptors.

We then performed a time-of-addition assay to determine how long the addition of EKL1C can be postponed before losing its antiviral activity in cell culture, in order to pinpoint which stage of the viral replication circle is targeted by EKL1C. For example, the HIV fusion-inhibitor-based antiviral drug T20 lost more than 50% of its antiviral activity when it was added to cells 2 h after the addition of HIV-1 [13]. As shown in Figure 3B, the inhibitory activity of EKL1C remained above 90% after 0.5 h of HIV-1 IIIB infection. However, when EKL1C was added 1 or 2 h after infection, its inhibitory activity against HIV-1 infection significantly waned. With the addition of time, the inhibitory activity of EKL1C decreased gradually to less than 20% at 4, 6, and 8 h after infection, indicating that like the HIV-1 fusion inhibitor T20, EKL1C inhibits HIV-1 entry at the virus–cell fusion step, the early stage of viral replication.

### 2.6. EKL1C Inhibited HIV-1-Env-Mediated Cell–Cell Fusion

HIV-1 entry into target cells is a multistep process involving virus attachment, membrane fusion, and entry [14,15]. We used MT-2 cells expressing receptor CD4 and co-receptor CXCR4 as target cells and H9/HIV-1_IIIB_ cells expressing HIV-1 Env as effector cells to evaluate the inhibitory activity of EKL1C against HIV-1-Env-mediated cell fusion. As shown in Figure 4A, EKL1C could effectively inhibit HIV-1-Env-mediated cell–cell fusion with an IC_50_ of 7 nM, which is 2.5-fold more potent than that of T20, confirming that EKL1C inhibits HIV-1 infection through the suppression of viral Env-mediated fusion between viral and target cell membranes.

### 2.7. EKL1C Bound to the HR1 of gp41 and Blocked the Formation of 6-HB

The EKL1C peptide was designed on the basis of EK1 and EK1C4, both of which can inhibit the formation of coronavirus 6-HB and effectively inhibit viral fusion, as proved in our previous studies [12,16,17]. Therefore, we next investigated if EKL1C could inhibit HIV-1 infection by targeting the HR1 domain of gp41 to block homologous 6-HB formation. According to previous studies, N36 and C34 could form 6-HB at an equimolar concentration and show a corresponding band on native polyacrylamide gel electrophoresis (N-PAGE) gel [18]. Consistent with the previous report [18], N36 showed no band because it containment negative charges that drove the peptide upward and off the gel, while C34 and EKL1C showed their bands at the lower and middle positions, respectively, in the gel. The mixture of N36/C34 showed a band at the top position in the gel, which correspond to the band of 6-HB, while the mixtures of N36 + C34 + EKL1C at the increasing concentration displayed 6-HB bands (upper part) with decreasing density and the C34 bands (lower part) with increasing density, respectively (Figure 4B), suggesting that EKL1C is able to block 6-HB formation between N36 and C34 in a dose-dependent manner.

### 2.8. EKL1C at 5 μM Exhibited no Obvious In Vitro Cytotoxicity to MT-2 Cells, CEMx174 517 5.25 M7 Cells, and U87 CD4^+^ CCR5^+^ Cells

We evaluated the in vitro safety of EKL1C by adding different concentrations of EKL1C to target cells (MT-2 cells, CEMx174 517 5.25 M7 cells, or U87 CD4^+^ CCR5^+^ cells) cultured at 37 °C with 5% CO_2_ for 48 h and detected cytotoxicity with a CCK8 kit. As shown in Figure 5, we found that the CC_50_ (50% cytotoxicity concentration) values of EKL1C to MT-2, M7, and U87 cells were 7.8, 17.3, and 32.3 μM, respectively. These results suggest that EKL1C has no obvious toxicity to cells at a concentration of 5 μM and the inhibitory activity of EKL1C on HIV-1 infection was not caused by its cytotoxicity.

## 3. Discussion

In this study, we used T20, the first HIV fusion inhibitory peptide-based antiviral drug, as a control to determine the sensitivity of the HIV-1 strains tested in this study to the HIV fusion inhibitor and for validation of the HIV-1 inhibition assays that were used to evaluate the inhibitory activity of the pan-HCoV fusion inhibitor EKL1C against HIV-1 infection. We found that EKL1C exhibited potent and broad-spectrum inhibitory activity against divergent HIV-1 strains, including HIV-1 laboratory-adapted strains, primary isolates, and T20-resistant strains. Mechanistically, EKL1C binds to the HR1 domain to block homologous 6-HB formation between the HR1 and HR2 domains in viral gp41, thereby inhibiting fusion between HIV-1 and cell membranes and, hence, infection, suggesting that the HR1 and HR2 regions involved in the formation of 6-HB of the class I enveloped viruses are conserved targets for the development of broad-spectrum inhibitors against class I enveloped viruses. Our previous studies have shown that peptides derived from the HR2 domain of the HIV-1 gp41, such as SJ-2176, C34, and T20, have been shown to be effective in inhibiting HIV-1 infection by competitively binding the exposed grooves on the viral gp41 HR1 trimer [19,20,21,22,23]. These results, as well as those reporting its inhibitory activity against a variety of HCoVs [11], suggest that EKL1C possesses broad-spectrum antiviral activity against HCoV, HIV-1, and possibly other class I enveloped viruses and has the potential for further development as a new antiviral drug.

We previously found that a variety of lipopeptide-based pan-CoV fusion inhibitors, such as EK1C4, could inhibit the infection of both HCoV and HIV-1 [9]. This suggests that the mechanism of HCoV fusion inhibitors is similar to that of the fusion inhibitors against other class I enveloped viruses, such as HIV-1. Unlike the PEGylated lipopeptide EK1C4, that may induce anti-PEG antibodies in patients who use EK1C4 [10,11], the dePEGylated lipopeptide EKL1C is expected to not elicit anti-PEG antibodies in vivo, thus having higher safety and stability and lower cost, and being easier to synthesize. Therefore, compared with EK1C4, dePEGylated lipopeptide EKL1C will be more suitable for long-term use in the treatment of HIV or HCoV infection in the future.

In the current COVID-19 pandemic, patients who are simultaneously infected with HIV and COVID-19 in South Africa and other regions, owing to their low immunity, may develop novel HCoV variants that escape the immune system. This calls for the development of bi-/multi-functional drugs that target the conserved target of HCoV HR1 and also have therapeutic effects on HIV. As a potent lipopeptide-based pan-CoV and pan-HIV-1 viral fusion inhibitor, EKL1C is expected to be a potential candidate against infection by HIV-1, HCoV, and the potential emerging and re-emerging class I enveloped viruses in the future.

## 4. Materials and Methods

### 4.1. Cells, Viruses, and Peptides

CEMx174 517 5.25 M7 cells, MT-2 cells, U87 CD4^+^ CCR5^+^ cells, HIV-1 IIIB chronically infected H9 (H9/HIV-1_IIIB_) cells, HIV-1 laboratory-adapted and primary HIV-1 isolates, as well as HIV-1 T20-resistant strains were obtained from the NIH AIDS Reagent Program. 293T cells were obtained from the American Type Culture Collection (ATCC, Manzas, VA, USA). Peptides T20 (YTSLIHSLIEESQNQQEKNEQELLELDKWASLWNWF), N36 (SGIVQQQNNLLRAIEAQQHLLQLTVWGIKQLQARIL), and C34 (WMEWDREINNYTSLIHSLIEESQNQQEKNEQELL) were synthesized by KareBay Biochem with a purity of >95% (Figure 1A).

EKL1 (NVTFLDLEYEMKKLEEAIKKLEESYIDLKELGTYEY) and EKL0C (SLDQINVTFLDLEYEMKKLEEAIKKLEESYIDLKELGTYEYYVKW-GSG-chol) were synthesized by Nankai Hecheng S&T Co., Ltd. (Tianjin, China) with a purity of >95%, while EKL1C (NVTFLDLEYEMKKLEEAIKKLEESYIDLKELGTYEY-GSG-chol) was synthesized by Dr. Chao Wang with a purity of >95%.

### 4.2. Plasmids

The HIV-1 envelope-expressing plasmids pcDNA3.1-REJO4541-Env, pcDNA3.1-BC02-Env, pcDNA3.1-CH119-Env, pcDNA3.1-GX11.13-Env, pcDNA3.1-ZM53M.PB12-Env, pcDNA3.1-HIV-25710-2 clone 43-Env, pcDNA3.1-CRF01_AE clone 269-Env, pcDNA3.1-Bal.01-Env, pcDNA3.1-SF162-Env, pcDNA3.1-JRFL-Env, pcDNA3.1-pRHPA4259 clone 7 (SVPB14)-Env, pcDNA3.1-CRF02_AG clone 266-Env, pcDNA3.1-CRF02_AG clone 271-Env, pcDNA3.1-CRF02_AG clone 278-Env, pcDNA3.1-Subtype G clone 252-Env, pcDNA3.1-Q769env.h5-Env, pcDNA3.1-Q842env.d16-Env, pcDNA3.1-Q259env.w6-Env, pcDNA3.1-QA013.70I.ENV.M12-Env, pcDNA3.1-QD435.100M.ENV.E1-Env, and pNL4-3.Luc.RE (Luciferase reporter vector) plasmid were constructed and maintained in our laboratory.

### 4.3. Inhibition of HIV-1 Pseudovirus Infection

HIV-1 pseudoviruses were generated as previously described [24]. Briefly, HEK293T cells with 40~60% cell density was co-transfected with HIV-1 envelope-expressing plasmids and pNL4-3.Luc.RE plasmid (Luciferase reporter vector) by using VigoFect transfection reagent (Vigorous Biotechnology, Beijing, China). Fresh DMEM with 10% FBS was replaced 6–8 h after transfection. HIV-1 pseudoviruses were harvested at 48 h post transfection and stored at −80 °C.

The HIV-1 pseudovirus inhibition assay was performed as described previously [25]. Briefly, 1 × 10^4^/well of U87 CD4^+^ CCR5^+^ cells 100 µL were cultured in a 96-well cell culture plate at 37 °C with 5% CO_2_ for 12–16 h. Peptides with different concentrations were mixed with a virus at 100 TCID_50_ and incubated at 37 °C for 30 min before adding U87 CD4^+^ CCR5^+^ cells. After 12 h of infection, fresh medium was replaced. After another 48 h of culture, target cells were lysed with Cell Culture Lysis (Promega, Madison, WI, USA) and cell lysate was added to a 96-well Costar plate (Corning Costar, NY, USA). Luciferase kit (Promega, Madison, WI, USA) and a multi-detection Microplate Reader (Ultra 384, Tecan) capable of reading luminescence was used to detect luciferase.

### 4.4. Inhibition of HIV-1 Infection

The inhibitory activity of peptides against HIV-1 laboratory-adapted strains IIIB (X4) and Bal (R5), HIV-1 primary isolates, and T20-resistant strains was detected as previously described [21,26]. Briefly, 50 μL of different concentrations of peptide in PBS or PBS (as control) mixed with 50 μL of HIV-1 at 100 TCID_50_ was incubated at 37 °C for 30 min. Then, 100 µL containing 4 × 10^5^/mL MT-2 (for X4 virus) or 2 × 10^5^/mL CEMx 174 5.25 M7 cells (for R5 virus) was added and incubated at 37 °C overnight. After 16 h, the medium was replaced with fresh RPMI-1640 medium containing 10% FBS, and cells were cultured for 3 to 4 days. Cytopathic changes were observed, followed by collecting 50 μL of cell supernatant and mixing with 5% Triton X-100. P24 antigen in supernatant was detected by ELISA.

ELISA assay was performed as described previously [27]. Briefly, 50 µL of 5 μg/mL Anti-HIV Immunoglobulin (HIVIG) was added to the ELISA plate overnight at 4 °C and 5% nonfat milk at 37 °C was added as blocking buffer for 2 h. Then, cell supernatant was collected, added to the ELISA plate, and incubated at 37 °C for 1 h. Anti-p24 mAb 183, rabbit Anti-Mouse IgG-HRP (DACO), and substrate 3,3,5,5-TMB (Sigma-Aldrich) were added sequentially. P24 antigen was measured at 450 nm (A450) using a multi-detection microplate reader (Ultra 384, Tecan). Calcusyn software (Biosoft) was used to calculate IC_50_, and GraphPad Prism (GraphPad Software) was used to draw the suppression curve.

### 4.5. Cell-Washout Assay

To determine whether EKL1C inhibits HIV-1 infection by acting on cells or not, we conducted the cell-washout assay as previously described [28]. Briefly, MT-2 cells (1 × 10^5^/mL) were incubated with EKL1C (500 nM) at 4 °C for 1 h and washed by centrifugation at 800 rpm for 5 min with serum-free RPMI-1640 medium 3 times to remove the unbound peptide, then HIV-1 IIIB was added to the washed MT-2 cells or non-washed cells (as control). At 12–16 h post infection, the culture supernatants were replaced with fresh RPMI-1640 medium containing 10% FBS. On the fourth day post infection, p24 antigen in supernatant was detected by ELISA as described above.

### 4.6. Time-of-Addition Assay

To determine whether EKL1C inhibits HIV-1 infection by acting at the viral entry stage, we performed the time-of-addition assay as previously described [28,29]. Briefly, 100 TCID_50_ HIV-1 IIIB was incubated with 1 × 10^5^/mL MT-2 cells for 0, 0.5, 1, 2, 4, 6, and 8 h at 37 °C before the addition of EKL1C (500 nM). At 2 h post infection, the culture supernatants were replaced with fresh RPMI-1640 medium containing 10% FBS. On the fourth day post infection, p24 antigen in supernatant was detected by ELISA as described above.

### 4.7. HIV-1-Env-Mediated Cell–Cell Fusion Assay

The inhibitory activity of peptides against HIV-1-Env-mediated cell–cell fusion was detected as described previously [30,31]. Briefly, 2 × 10^5^/mL H9/HIV-1_IIIB_ cells were labeled with 2.5 µL 1 mM fluorescent reagent, Calcein AM (Themo Fisher Invitrogen, USA), and incubated at 37 °C for 30 min. H9/HIV-1_IIIB_ cells were washed twice with PBS by centrifugation at 800 rpm for 3 min and resuspended to 2 × 10^5^/mL in fresh RPMI-1640 medium containing 10% FBS. An amount of 50 µL of peptide at indicated concentrations was added to wells of a 96-well cell culture plate, and 50 µL of labeled H9/HIV-1_IIIB_ cells was added to each well, followed by incubation at 37 °C for 30 min. Then, 100 µL of 1 × 10^6^/mL MT-2 cells were added to each well and incubated at 37 °C for 2 h. Cell–cell fusion of each well was photographed by inverted fluorescence microscopy (Zeiss, Oberkochen, Germany). The wells containing H9/HIV-1_IIIB_ cells and MT-2 cells only were used as positive controls, while those containing H9/HIV-1_IIIB_ cells only were used as negative controls. Calcusyn software (Biosoft) was used to calculate the inhibition of cell–cell fusion and IC_50_, and GraphPad Prism (GraphPad Software) was used to present the suppression curve.

### 4.8. N-PAGE

N-PAGE was performed to determine the potential inhibitory activity of EKL1C on 6-HB formation between HIV-1 HR1 peptide N36 and HR2 peptide C34 as previously described [32]. Briefly, the mixture of C34 + PBS, N36 + PBS, EKL1C + PBS, and N36 (50 µM) was incubated with EKL1C at 0, 20, or 40 µM, respectively, at 37 °C for 30 min, followed by addition of C34 (20 µM) and incubation at 25 °C for 10 min. In the controls, C34, N36, and EKL1C were incubated with PBS at 37 °C for 30 min and 25 °C for 10 min. Tris-glycine native sample buffer was added to each of the above samples at a ratio of 1:5, and the mixtures were loaded onto 18% Tris-glycine gel for gel electrophoresis at room temperature at a constant voltage of 125 V for 180 min. The gel was stained with Coomassie Blue and imaged with a FluorChem 8800 imaging system (Alpha Innotech Corp., San Leandro, CA, USA).

### 4.9. Cytotoxicity Assay

Cytotoxicity of peptide EKL1C to cells was assessed as previously described [33]. Briefly, 50 μL of EKL1C at different concentrations, together with 100 µL of MT-2 cells, CEMx174 517 5.25 M7 cells, or U87 CD4^+^ CCR5^+^ cells (3 × 10^4^ cells/well), was cultured in wells of a 96-well cell culture plate at 37 °C with 5% CO_2_ for 48 h. Then, 50 µL Cell Counting Kit-8 (CCK-8; Dojindo, Kumamoto, Japan) was diluted 5× and added, followed by an additional incubation at 37 °C for 4 h. The absorbance was measured at 450 nm by using the Ultra 384 microplate reader (Tecan). Calcusyn software (Biosoft) was used to calculate CC_50_ of EKL1C, and GraphPad Prism (GraphPad Software) was used to draw the histogram.

## Figures and Tables

**Figure 1 ijms-24-09779-f001:**
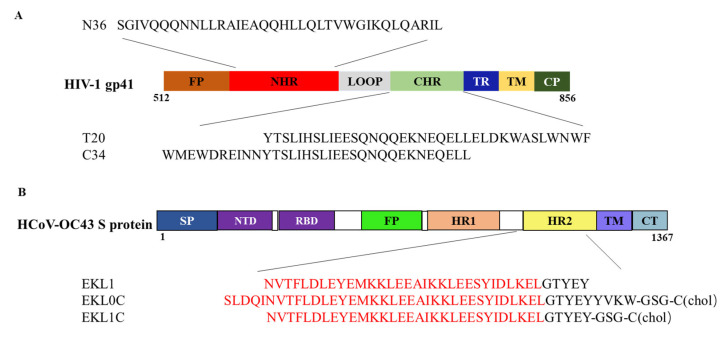
Structure of viral envelope protein and sequence of peptides: (**A**) Schematic diagram of HIV-1 gp41 and sequences of peptides derived from the gp41 HR1 and HR2. Fusion peptide region, FP; N-terminal heptad repeat, HR1; C-terminal heptad repeat, HR2; tryptophan-rich region, TR; transmembrane region, TM; cytoplasm region, CP. N36 derived from HR1 domain and C34 and T20 derived from HR2 domain are shown in the diagram. (**B**) Schematic diagram of HCoV spike (S) protein and sequences of peptides derived from S protein HR2 domain. Signal peptide, SP; N-terminal domain, NTD; receptor-binding domain, RBD; fusion peptide, FP; heptad repeat 1, HR1 domain; heptad repeat 2, HR2 domain; transmembrane domain, TM; and cytoplasmic domain, CP. EKL1, EKL0C, EKL1C derived from HR2 domain are shown in the diagram.

**Figure 2 ijms-24-09779-f002:**
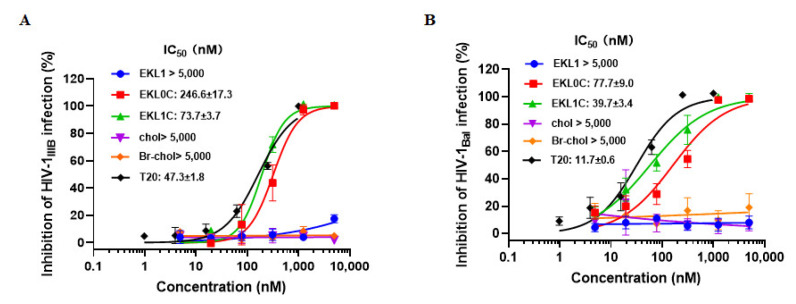
Inhibitory activity of EKL1C against HIV-1 infection: (**A**) Inhibitory activity of EKL1, EKL0C, EKL1C, chol, Br-chol, and T20 against infection from HIV-1 IIIB (subtype B, X4). (**B**) Inhibitory activity of EKL1, EKL0C, EKL1C, chol, Br-chol, and T20 against infection from HIV-1 Bal (subtype B, R5). Each sample was tested in triplicate, the experiment was repeated two to three times, and the data are presented as mean ± SD.

**Figure 3 ijms-24-09779-f003:**
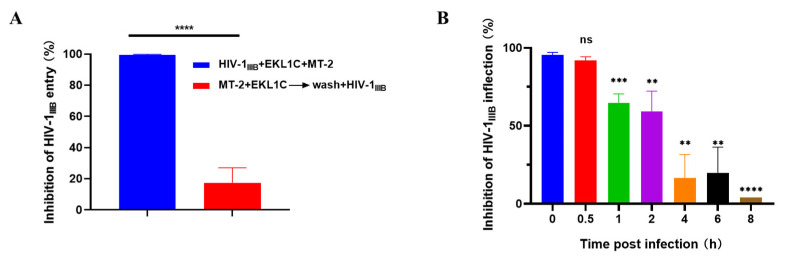
EKL1C inhibited HIV-1 infection at the early stage of viral replication. (**A**) EKL1C inhibits virus entry not by interacting with cell surface proteins as determined by cell-washout assay. The inhibitory activity of EKL1C on HIV-1_IIIB_ entry was detected after the EKL1C-treated MT-2 cells were washed (red bar) or not washed (blue bar) by centrifugation. Each sample was tested in triplicate, and the experiment was repeated two to three times. Data from a representative experiment are presented as mean ± SD as means ± SD, ****, *p* < 0.0001. (**B**) Inhibition of HIV-1 entry at the early stage of viral replication by EKL1C as determined by time-of-addition assay. Each sample was tested in triplicate, and the experiment was repeated two to three times. Data from a representative experiment are presented as mean ± SD. ns, no significance, **, *p* < 0.01, ***, *p* < 0.001, ****, *p* < 0.0001.

**Figure 4 ijms-24-09779-f004:**
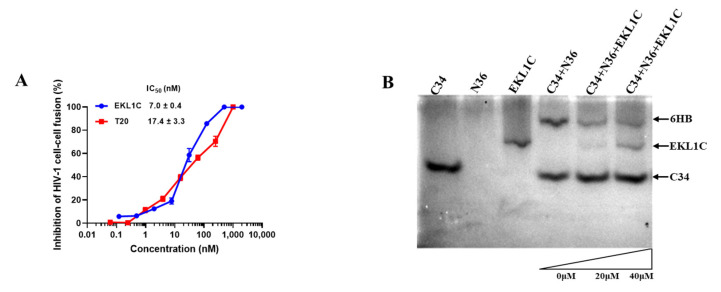
The mechanism of EKL1C peptide against HIV-1 infection: (**A**) Inhibition of Env-mediated cell–cell fusion between H9/HIV-1IIIB cells and MT-2 cells. Each sample was tested in triplicate, and the experiment was repeated two to three times. Data from a representative experiment are presented as mean ± SD. (**B**) N-PAGE showed that EKL1C blocked the formation of 6-HB between N36 and C34.

**Figure 5 ijms-24-09779-f005:**
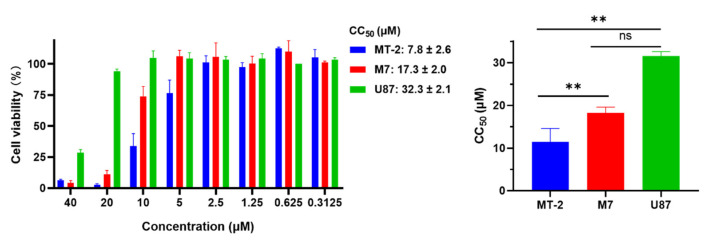
The potential cytotoxicity of EKL1C to MT-2, M7, and U87 cells. Each sample was tested in triplicate and the experiment was repeated two to three times. Data from a representative experiment are presented as mean ± SD. ns, no significance, **, *p* < 0.01.

**Table 1 ijms-24-09779-t001:** Inhibitory activity of EKL1C and T20 against HIV-1 pseudovirus.

Virus	Titer	IC_50_ (nM)	
		EKL1C	T20
Q769env.h5 (A, R5)	2	22.1 ± 14.4	179.3 ± 76.9
Q842env.d16 (A, R5)	2	51.2 ± 12.7	23.8 ± 30.3
Q259env.w6 (A, R5)	2	39.0 ± 33.9	50.8 ± 22.9
Bal (B, R5)	3	37.6 ± 26.2	23.2 ± 15.8
REJO4541 (B, R5)	-	17.6 ± 15.8	11.6 ± 13.7
SF162 (B, R5)	1A	186.5 ± 85.3	302.7 ± 164.1
JRFL (B, R5)	2	158.4 ± 57.9	222.4 ± 170.3
pRHPA4259 clone 7 (B, R5)	2	147.4 ± 64.2	14.5 ± 5.9
ZM53M.PB12 (C, R5)	2	6.6 ± 6.8	51.4 ± 41.0
HIV-25710-2, clone 43 (C, R5)	1B	23.8 ± 20.6	53.0 ± 27.1
QA013.70I.ENV.M12 (D, R5)	2	31.5 ± 13.8	6.5 ± 3.5
QD435.100M.ENV.E1 (D, R5)	2	48.1 ± 33.1	72.0 ± 24.8
Subtype G clone 252 (G, R5)	2	98.5 ± 59.4	27.5 ± 9.5
CRF01_AE clone 269 (A/E, R5)	-	5.1 ± 1.234	14.2 ± 17.5
GX11.13 (A/E, R5)	-	10.3 ± 7.5	74.1 ± 60.2
CRF02_AG clone 266 (A/G, R5)	2	37.5 ± 3.3	37.1 ± 7.2
CRF02_AG clone 271 (A/G, R5)	1B	19.1 ± 8.2	11.5 ± 6.4
CRF02_AG clone 278 (A/G, R5)	3	19.4 ± 14.2	28.4 ± 28.6
BC02 (B/C, R5)	-	20.5 ± 11.0	31.7 ± 34.0
CH119 (B/C, R5)	2	74.2 ± 45.6	394.2 ± 71.6

Note: Each sample was tested in triplicate and the experiment was repeated two to three times. Data from a representative experiment are presented as mean ± SD.

**Table 2 ijms-24-09779-t002:** Inhibitory activity of EKL1C and T20 against HIV-1 clinical isolates.

Virus	Subtype, Tropism	IC_50_ (nM)	
		EKL1C	T20
MN/H9 (84US_MNp)	(A, X4)	98.1 ± 29.2	52.6 ± 7.4
BZ167/GS 010 (89BZ_167)	(B, X4)	118.4 ± 17.0	47.3 ± 5.9
92UG024	(D, X4)	80.2 ± 7.6	31.0 ± 5.6
J32228M4	(D, R5)	70.1 ± 3.2	26.2 ± 3.1
HIVBCF02	(O, R5)	192.9 ± 37.6	41.3 ± 1.4

Note: Each sample was tested in triplicate, and the experiment was repeated two to three times. Data from a representative experiment are presented as mean ± SD.

**Table 3 ijms-24-09779-t003:** Inhibitory activity of EKL1C and T20 against T20-resistant HIV-1 strains.

Virus	IC_50_ (nM)	
	EKL1C	T20
V38A, N42D	36.1 ± 11.2	>1000
HIV-1 NL4-3 D36G (WT)	17.2 ± 10.2	10.2 ± 4.1
V38A	97.9 ± 28.3	>1000
N42T, N43K	9.4 ± 1.2	>1000
V38E, N42S	19.7 ± 10.3	>1000
V38A, N42T	17.4 ± 1.5	>1000

Note: Each sample was tested in triplicate, and the experiment was repeated two to three times. Data from a representative experiment are presented as mean ± SD.

## Data Availability

Not applicable.

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
