# Peer review of "A dePEGylated Lipopeptide-Based Pan-Coronavirus Fusion Inhibitor Exhibits Potent and Broad-Spectrum Anti-HIV-1 Activity without Eliciting Anti-PEG Antibodies"

_ijms, 2023, doi:10.3390/ijms24119779_

Round 1
Reviewer 1 Report
Ling Xu and coll show the activity of a dePEGylated lipopeptide-based pan-coronavirus fusion inhibitor versus a broad range of HIV ENV pseudotypes. The experiments have been well organized and properly carried out Nevertheless I would like to rise some points.
Major
Table 1. The authors sustain in the result section “The dePEGylated lipopeptide-based pan-CoV fusion inhibitor EKL1C exhibited potent inhibitory activity against a broad spectrum of pseudotyped HIV-1 strains” that EKL1C has a broad range of activity versus HIV ENV pseudotype. The results shown in the Table derived from values in triplicate and therefore it is difficult to assess how significant they are in respect to the values obtained with T20. This is a crucial aspect to be investigated and therefore the authors should provide more replicate in order to perform a statistic evaluation.
Fig 2. The authors should more clearly explain the difference among the 3 lipopeptides EKL1, EKL0C and EKL1C tested versus HIV laboratory strain. Actually in fig 2 it is shown a remarkable variability in the inhibitory activity in HIVIIIB in respect to HIV Bal .
Table 2. Unfortunately EKLIC is less potent than T20 toward clinical isolates. The authors should comments this in the discussion section.
Figure 4. The figure provided is not clear. The authors should try to obtain a better images from the original one they have provided.
Figure 5. The cytotoxic activity shown on three cell lines derive from experiment in triplicate. It is unclear how many experiments the authors have performed and also in this case it is necessary to calculate statistically the significance of the difference within the obtained data. Therefore more experiments have to be performed.
Minor: The authors have to correct the several English spelling errors and revise truly the English form.

The english grammar and form have to be revised
Reviewer 2 Report
In the manuscript written by Ling et.al, the authors have estimated the inhibitory activity of dePEGylated lipopolypeptide EKL1C against HIV strains. The results are encouraging but the manuscript needs improvement and some of the assays are required to be repeated.
Following suggestions could be considered:
The authors could have used the positive control to compare the results of EKL1C and T20 inhibitory activity. That way, the authors could deduce the the efficiency of potent inhibitors.
As authors have talked about the inhibitory activity of chosen lipopeptides against coronavirus, authors could have provided reference to that respective article and compared the previous results to the current one in Table1.
In Table 2, the EKL1C did show higher activity against other clinical isolates but not J32228M4, it would be helpful for the readers, if authors could elaborate or mention this in text and could comment on it.
In Figure 2.5 B authors could comment in detail about the diminishing HIV inhibition by EKL1C after 1, 2, 4, 6 h of infection. The data needs to be analyzed statistically by post hoc analysis to determine the significance of variance. The -0.5 time needs to be removed from the graph, as it makes difficult to comprehend that how could the inhibition be determined before the infection with HIV. Figure legends should also be modified respectively.
In Figure 4 the PAGE is not quite convincing and could be repeated.
In Figure 5, the statistical analysis comparing the different groups could be added. The description of cell toxicity analysis could be more detailed, including the incubation hours of cells with the ELIC and other peptides.
Please mention under results, for how long the cells were incubated with inhibitors.
Please add references for lines 213, 214.
Conclusion needs to be added.
Needs improvement
Round 2
Reviewer 1 Report
THe manuscript has been correctly improved and therefore as, it is, is suitable for publication
Reviewer 2 Report
Authors have included most of the suggest changes.
Minor changes in English language